# Further Development of SAMPDI-3D: A Machine Learning Method for Predicting Binding Free Energy Changes Caused by Mutations in Either Protein or DNA

**DOI:** 10.3390/genes16010101

**Published:** 2025-01-19

**Authors:** Prawin Rimal, Shamrat Kumar Paul, Shailesh Kumar Panday, Emil Alexov

**Affiliations:** Department of Physics and Astronomy, College of Science, Clemson University, Clemson, SC 29634, USA; primal@clemson.edu (P.R.); shamrap@clemson.edu (S.K.P.); spanday@clemson.edu (S.K.P.)

**Keywords:** protein–DNA binding, binding free energy, machine learning, protein mutations, DNA mutations, database of binding free energy changes

## Abstract

Background/Objectives: Predicting the effects of protein and DNA mutations on the binding free energy of protein–DNA complexes is crucial for understanding how DNA variants impact wild-type cellular function. As many cellular interactions involve protein–DNA binding, accurately predicting changes in binding free energy (ΔΔG) is valuable for distinguishing pathogenic mutations from benign ones. Methods: This study describes the development and optimization of the SAMPDI-3Dv2 machine learning method, which is trained on an expanded database of experimentally measured ΔΔGs. This enhanced model incorporates new features, including the 3D structure of the mutant protein, features of the mutant structure, and a position-specific scoring matrix (PSSM). Benchmarking was conducted using 5-fold cross-validation. Results: The updated SAMPDI-3D model (SAMPDI-3Dv2) achieved Pearson correlation coefficients (PCCs) of 0.68 for protein and 0.80 for DNA mutations. These results represent significant improvements over existing tools. Additionally, the method’s rapid execution time enables genome-scale predictions. Conclusions: The improved SAMPDI-3Dv2 shows enhanced predictive performance for analyzing mutations in protein–DNA complexes. By leveraging structural information and an expanded training dataset, SAMPDI-3Dv2 provides researchers with a more accurate and efficient tool for mutation analysis, contributing to identifying pathogenic variants and improving our understanding of cellular function.

## 1. Introduction

Protein–DNA interactions (PDIs) are crucial for the regulation of genetic information and integrity of cellular functions [1,2]. Transcription factors (TFs) are a specialized class of proteins that use their DNA-binding domains to regulate gene expression [3]. PDIs also regulate cellular processes inside cells, such as transcription, DNA repair, chromatin remodeling, cell cycle control, apoptosis, immune response, and epigenetic control [4,5]. In a protein–DNA complex, a point mutation either in the protein (missense mutation) or the cognate DNA impacts binding affinity and specificity, which is computed as a change in the binding free energy—ΔΔG = ΔG(mutant) − ΔG(wild-type), depicting the differences in binding free energy between the mutant and wild type. A point mutation may disrupt the normal function of cells and may cause diseases such as cancer [6], Alzheimer’s [7], cardiovascular disease [8], and neurological disorders [9,10].

Thus, to understand the pathogenic potential and develop therapeutic strategies, the precise quantification of the impact of mutations on PDIs is essential. Numerous experimental methods are available to estimate the impact of mutation on PDIs. Among these, the electrophoretic mobility shift assay (EMSA), based on the idea that the electrophoretic mobility of free DNA is higher than that of a protein–DNA complex, is widely used [11]. Free DNA [12] migrates faster through the gel matrix during electrophoresis because of its higher charge/mass ratio. The differences in electrophoretic mobility between DNA bound to a protein and free DNA are useful to quantify the dissociation constant (*K*_d_), which serves as a determinant of binding affinity [13,14]. A lower *K*_d_ value indicates stronger binding between the protein and DNA. Isothermal titration calorimetry (ITC) is another technique that provides thermodynamic parameters including the reaction enthalpy and binding stoichiometry of the protein–DNA complex by titrating a protein solution into a DNA solution [15,16]. The heat changes associated with each batch of protein solution in the titration reaction are then integrated and plotted against the molar ratio of the interacting DNA molecules to generate the binding isotherm [17]. This isotherm is subsequently analyzed using nonlinear least squares (NLLS) fitting to obtain the binding affinity (*K*_d_) of the interacting protein and DNA [17]. Surface plasmon resonance (SPR) is a label-free technique for studying protein–DNA interactions. This method involves immobilizing DNA on a sensor surface and measuring the changes in the refractive index caused by binding events when a solution containing the protein flows over the surface [18,19]. The changes in the refractive index affect the resonance conditions of the surface plasmon wave, leading to a quantifiable shift in the angle at which the light is reflected, producing the SPR signal [20]. This SPR signal is plotted over time, producing a sensorgram showing the interaction’s kinetics. By analyzing the sensorgram, the association and dissociation rates of protein and DNA can be used to calculate the equilibrium dissociation constant (*K*_d_), which reflects the binding affinity of the protein–DNA complex [21]. However, these traditional methods cannot be applied to large-scale investigations because they require substantial sample amounts (SPR, nanomoles; EMSA, micromoles; ITC, tens of micromoles) [22] and a long time to deliver the results.

In addition to traditional methods, high throughput methods can be used to investigate the impacts of mutations on PDIs. For example, the protein-binding microarray (PBM) is a fluorescence-based method that assesses the binding specificities of transcription factors (TFs) to interacting DNA [21,23]. PBM measures the fluorescence intensity, which indicates the amount of TF bound to the DNA and provides the binding affinity and dissociation constant (*K*_d_) parameters. The high-throughput systematic evolution of ligands by exponential enrichment (HT-SELEX) combines the traditional SELEX with high-throughput sequencing to determine the binding specificities of DNA-binding proteins [24]. This is achieved by recognizing the preferred DNA-binding motifs of a TF by selecting DNA sequences from a random oligonucleotide library, purifying the protein–DNA complexes, and amplifying the bound DNA through a polymerase chain reaction (PCR), which allows for quantifying the affinity between the protein and the interacting DNA [24,25]. Recently developed methods determine the relative binding affinity of TFs by integrating EMSA and HT-SELEX data. For example, the No Read Left Behind (NRLB) method [26] employs biophysical methods and statistical models to predict protein–DNA binding affinity from single-round SELEX data. Another study [27] used a similar combination of EMSA and HT-SELEX techniques to study the binding mechanism of TFs. Moreover, the binding specificity of a TF can be estimated by applying the BEMSER method, which uses data from protein-binding microarrays (PBM) [28]. Although the above-mentioned methods provide essential data regarding protein–DNA binding affinity, they depend on techniques such as PBM, EMSA, or HT-SELEX; thus, only a quantification of binding affinity in relative terms is attained, which cannot predict ΔΔG [29]. Hence, traditional methods should be used to measure the binding affinity of protein–DNA complexes, even though they are labor-intensive and expensive for high-throughput measurements [29].

The need for quantifying binding affinity data across a genomic scale has encouraged the development of computational techniques. Our previously developed method, SAMPDI, is based on the enhanced MM/PBSA method, combining molecular mechanics energy calculations and continuum solvation models alongside knowledge-based descriptors to predict changes in protein–DNA binding free energy caused by single-point protein mutations [30]. However, SAMPDI could not predict the impact of DNA mutations. Therefore, we developed SAMPDI-3D by employing a gradient-boosting decision tree algorithm, thus incorporating a wide range of features including physicochemical properties, structural characteristics of the mutation site, and protein–DNA interactions. This allows the model to predict changes in binding free energy resulting from both single-point protein and DNA mutations [29].

mCSM-NA [31] and mmCSM-NA [32] are other models available and are primarily built upon graph-based structural signatures to predict the impact of mutations in proteins interacting with DNA/RNA [31,32]. mCSM-NA emphasizes single-point missense mutations, depicting protein and nucleic acid structures as a graph while atoms and edges represent nodes reflecting interactions [31]. mCSM-NA introduces the desired mutation by altering the graph-based signature of the wild-type residue, thus introducing changes in the pharmacophore modeling and physicochemical properties. Then, it compares the altered graph to the wild-type graph to quantify changes in binding affinity (ΔΔG) [31]. mmCSM-NA expands on the same concept as mCSM-NA but can predict the effects of both single- and multiple-point mutations. This is because the model considers changes in protein stability, dynamics, non-covalent interactions, and residue depth. Thus, mmCSM-NA is an optimized version of graph-based signatures and uses a similar approach to that of mCSM-NA to introduce multiple-point mutations into the protein to quantify the changes in binding affinity [31,32]. Another method, PremPDI, applies energy minimization and side-chain optimization algorithms to compare the interaction energies of the mutant and wild-type complexes to predict changes in binding free energy [33]. A recently developed algorithm named protein-nucleic acid binding affinity change estimator (PNBACE) [34] can predict the changes in binding affinity due to point and multiple mutations, either in the protein or in a nucleic acid (DNA/RNA) in the protein–nucleic acid complex. Decomposing the binding free energy of a complex into pairwise interaction energies between atoms provides the overall energetic landscape of the interactions within the complex. The algorithm builds different energy networks from these obtained pairwise interactions and formulates energy-based topological features, along with partition-based energy features that depict the energy contribution of different complex segments. These features are then input to train individual machine learning (ML) models, which are then combined to produce an ensemble model, using a differential evolution algorithm, which ultimately predicts the impact of binding affinity due to the imposed mutation in the complex [34].

However, these aforementioned computational methods (mCSM-NA [31], mmCSM-NA [32], PremPDI [33], SAMPDI [30], SAMPDI-3D [29], and PNBACE [34]) were developed and trained on a few cases. This may limit their ability to accurately predict the impact of new mutations (i.e., mutation types unavailable in the training dataset). Furthermore, except for SAMPDI-3D and PNBACE, currently none of the other methods mentioned above can predict the effects of DNA mutation on ΔΔG. Moreover, PNBACE requires high computational time; thus, it cannot be applied to large-scale investigations.

The above-mentioned computational methods use a database of protein or DNA mutations and associated changes in binding free energy (ΔΔG) due to the mutation. These methods learn the relationship between the features and ΔΔG during the training phase and make predictions based on the learned associations. However, these databases are often very small, limiting the method’s generalizability, as pointed out in a previous work [35].

Therefore, we tested SAMPDI-3D and the other methods previously mentioned on the new entries in ProNAB to determine the limitations of existing predictors trained on a small database [36]. However, as shown in the Section 3.2, none of them performed well. This motivated us to develop SAMPDI-3D based on the newly available data points and add new features to address the identified limitations. This was achieved by developing a new version of SAMPDI-3D, named SAMPDI-3Dv2, which uses an approximately 42% larger dataset for protein mutations and a 9% larger dataset for DNA mutations in PDIs. In addition, we introduced new features, which resulted in better performance, as indicated by an improved Pearson correlation coefficient (PCC).

## 2. Materials and Methods

### 2.1. Data Cleaning (ProNAB)

There are 20,090 data points in the original ProNAB dataset [36]. However, many of these entries cannot be used because of insufficient information or typos. Thus, extensive manual cleaning of the original database was necessary, as outlined in Figure 1. Developing the model required experimentally measured changes in binding free energy (ΔΔG) upon mutation of either protein or DNA. Therefore, we deleted entries without the ΔΔG value, reducing the original dataset to 6661 data points. Next, we excluded data points related to RNA mutations (as SAMPDI-3D is aimed at protein–DNA binding free energy predictions), leaving 4290 data points. Furthermore, we discarded data points where protein and DNA mutations were mentioned simultaneously because our aim was only single-point mutations. Thus, the dataset was reduced to 3671 data points. Afterward, entries where changes in binding free energy resulted from two or more mutations in a single entry were filtered out, further refining the dataset to 3216 data points. Subsequently, we deleted entries that lacked the corresponding wild-type PDB ID, resulting in a final set of 646 data points, with point mutations in either DNA or protein. Of these, 551 represented protein mutations, and 95 represented DNA mutations.

Among the 551 protein mutations, we identified 148 protein mutations that overlapped with our previous SAMPDI-3D training dataset (S419), leaving us with 403 new protein mutation cases from ProNAB. ProNAB also provided the cognate DNA sequence information for protein mutation cases. For a given mutation, we extracted the DNA sequence from the corresponding PDB file and used Clustal Omega [37] to perform pairwise sequence alignment between the DNA sequences given in ProNAB and the corresponding PDB files. Upon sequence alignment, we discarded data points where sequence identity was below 80%, yielding 289 protein mutation data points with high DNA sequence identity (>80%) between ProNAB and the PDB file.

Multiple copies of protein and DNA sequences are sometimes present in a PDB file. For example, a protein chain may be labeled “A”, where the mutation is located, with DNA chains labeled “B” and “C”. Several copies of the protein and binding DNA are frequently observed due to crystallization and structure-solving artifacts. The binding free energy measurements are representative of the biological assembly of the complex. Therefore, we cleaned the PDB structure to keep only a single copy of the protein–DNA complex and called it “cleaned PDB”. The cleaned PDBs were generated from the original PDBs by keeping only a single copy of the interacting chains of protein and DNA, using UCSF Chimera [38]. Two polymer (protein/DNA) chains were considered to interact when one or more pairs of atoms from the two were within a distance threshold of 5.0 Å. Henceforth, the cleaned PDBs were used. Furthermore, among these 289 protein mutation data points, PDB files associated with 48 of them corresponded to very low-resolution structures that had only α-carbons in the protein backbone, preventing secondary structure determination and, in turn, ΔΔG prediction. Thus, we discarded these data points with structural issues and were left with 241 protein mutation data points.

Among the 241 remaining protein mutation data points, we identified 52 cases where the same mutation had different ΔΔG values. For example, mutation R52S in PDB 1AAY had three ΔΔG values. This left us with 189 data points. Thus, we categorized these 52 data points based on the standard deviation of their corresponding multiple ΔΔG for the same mutation. Upon the calculation of the standard deviation (SD), we removed two data points where SD was greater than 1 kcal/mol and two additional points with high ΔΔG values. Hence, we were left with 48 data points having SDs of less than 1 kcal/mol, for which we averaged their ΔΔG values, resulting in 11 unique data points. By merging these 11 unique data points with the remaining 189, we arrived at 200 new protein mutation datasets. However, we found that, out of these 200 data points, 11 cases for a particular PDB (1LMB chain 3) had residues mismatched in PDB, and another 12 cases overlapped with our old SAMPDI-3D protein mutation dataset S419. Thus, we discarded them and ended up with 177 new protein mutation data points. We named this dataset S177. Including the S419 dataset, consisting of 419 point mutations in the protein, derived from our previous version of SAMPDI-3D [29], we obtained a dataset of 596 point mutations in protein, which we henceforth call S596.

Regarding DNA mutations, we initially had 95 DNA mutation data points. We found that four of these data points were already present in our previous SAMPDI-3D D463 DNA mutation dataset, while three mutations were related to the U base, indicating RNA rather than DNA mutations. Additionally, we removed 34 data points that had issues with the PDB components (e.g., missing residues), resulting in 63 DNA mutation data points. Upon further observation, 21 cases failed because of a wild-type base mismatch at the mutation site or because the mutation site itself was missing in the associated PDB. Finally, after data cleaning, we found 42 new DNA mutation entries, which we named the D42 dataset. In addition, we had 460 data points from the D463 dataset compiled in our previous version of SAMPDI-3D; here, three mutation sites were missing in PDB and were thus ignored. By merging the dataset of 460 (old) and 42 new single-base/base-pair DNA mutations, we created a dataset consisting of 502 data points, henceforth called the D502 dataset.

These cleaned datasets, S177 and D42, can be downloaded from http://compbio.clemson.edu/SAMPDI-3Dv2/ (accessed on 26 December 2024).

### 2.2. Training Dataset for Protein Mutations

Our previous version of SAMPDI-3D used the S419 dataset for training and testing of ΔΔG caused by protein mutations. The S419 dataset consists of 419 single mutations in 96 proteins and was prepared by merging the S219 and S200 datasets from PremPDI [33] and the literature [29]. The data in S219 originally came from ProNIT [39] and dbAMEPNI [40]. On the other hand, the data in S200 were obtained from the literature at the time of SAMPDI-3D development and were not included in the ProNIT database.

For our development of SAMPDI-3D v2 in 2024, we curated protein mutation data from the ProNAB [36] database. The data were cleaned by following the aforementioned ProNAB data-cleaning process. We included 177 (labeled as S177) new single-protein mutation data points together with our previous 419 protein mutation data points (S419), which resulted in 596 combined protein mutation data points. The final dataset was named S596.

### 2.3. Training Dataset for DNA Mutations

For the previous version of SAMPDI-3D, the D463 dataset was constructed as the training dataset for DNA mutations. The data came from ProNIT and the literature available at the time of the publication of the previous version. For the DNA mutations, our training set combined the ProNIT database and data from the recent literature. The training set comprised 245 single mismatches and 218 single base-pair substitutions—a total of 463 mutations in 30 proteins with quantitatively characterized ΔΔGs. Among them, 123 were taken from the ProNIT database. This dataset was named D463.

To develop our SAMPDI-3D v2, we looked into the D463 dataset [29] and found that three PDBs did not have a mutation position. Thus, we eliminated these entries and constructed a D460-modified DNA mutation dataset. After several ProNAB cleaning operations, we ended up with 42 DNA mutation data points and created a new DNA mutation dataset called D42. Thus, the combined modified DNA mutation dataset from SAMPDI-3D (D460) and the recently curated DNA mutation dataset from ProNAB yielded the new DNA mutation dataset for our SAMPDI-3D v2, named D502.

### 2.4. Key Features in the SAMPDI-3D v2 Machine Learning Model

#### 2.4.1. Protein Mutation

A wide range of features were used to develop a machine learning model to predict changes in binding free energy in protein–DNA complexes caused by single amino acid mutations. These include evolutionary preferences (point I), physicochemical properties (points II and III), structural features (points IV, V, and VI), secondary structure preferences (point VII), protein–DNA interaction features (point VIII), and mutation-induced interaction perturbations (point IX). The definitions and calculations of these features are detailed below.

(I)Position-specific scoring matrix (PSSM)

The protein sequence was derived from the input protein–DNA complex structure. Initially, residues were identified from the coordinate records and complemented with missing residues listed in the “REMARK 465” record in the PDB header. These sets of residues were merged and aligned with the “SEQRES” record to ensure consistency. Any non-standard residues identified using the “MODRES” record were reverted to their corresponding standard residues. Expression tag residues, parsed from the “SEQADV” record, were excluded. Finally, residues were mapped to their one-letter codes, and the protein sequence was saved in the FASTA format.

The cleaned protein sequence was queried against the UniRef50 database [41], using PSI-BLAST (v2.10.0) [42], with default parameters for three iterations to identify homologous sequences and generate the corresponding PSSM, as described in SAAMBE-SEQ [43]. This generated a PSSM matrix, which was written as an ASCII text file. The matrix dimensions were n × 20, where n is the sequence length and the 20 columns represent the standard amino acids.

The following features were derived from the PSSM matrix.

  (a)Evolutionary sequence composition features: For each of the 20 amino acids, a vector of normalized odds ratios across all positions in the sequence was computed using f(x) = 1/(1 + e^−x^). The mean of these normalized odds ratios for each amino acid served as a feature, capturing its sequence composition preference.  (b)Evolutionary odds of the mutation: Calculated as the difference in odds ratios between the mutant and wild-type residues at the mutation site.

(II)Mutation type-related features

Net volume: The change in residue volume due to the mutation was computed as the difference in molar volumes of the mutant and wild-type residues.

Net hydrophobicity: The hydrophobicity difference between mutant and wild-type residues derived from Moon’s hydrophobicity index [44].

Net flexibility: The change in rotamer counts, calculated as the logarithmic difference between the rotamers of mutant and wild-type residues, using data from the Dunbrack rotamer library [45].

(III)Amino acid category features

Mutation hydropathy class. This is a categorical feature for which the 20 standard amino acids are grouped into three classes: glycine (G), histidine (H), proline (P), serine (S), threonine (T), and tyrosine (Y) as hydropathically neutral; aspartate (D), glutamate (E), lysine (K), asparagine (N), glutamine (Q), and arginine (R) as hydrophilic; and alanine (A), cysteine (C), phenylalanine (F), isoleucine (I), leucine (L), methionine (M), valine (V), and tryptophan (W) as hydrophobic [46]. Furthermore, based on the wild-type and mutated amino acid classes, a unique label is calculated as HCI(wild-type) × 3 + HCI(mutant), where HCI(x) gives the hydrophobicity-class-index of the amino acid x. Thus, all possible 20 × 20 combinations of wild-type and mutant amino acids are mapped to nine (3 × 3) different labels.

Mutation polarity class. This is a categorical feature for which the 20 standard amino acids are grouped into four classes: alanine (A), cysteine (C), phenylalanine (F), isoleucine (I), leucine (L), methionine (M), valine (V), tryptophan (W), glycine (G), and proline (P) as nonpolar; aspartate (D) and glutamate (E) as polar-acidic; lysine (K), arginine (R), and histidine (H) as polar-basic; and asparagine (N), glutamine (Q), serine (S), threonine (T), and tyrosine (Y) as polar-neutral [46]. Furthermore, based on the wild-type and mutated amino acid classes, a unique label is calculated as PCI(wild-type) × 4 + PCI(mutant), where PCI(x) gives the polarity-class index of the amino acid x. Thus, all possible 20 × 20 combinations of wild-type and mutant amino acids are mapped to 16 (4 × 4) different labels.

Mutation size class. This is a categorical feature for which the 20 standard amino acids are grouped into five classes: alanine (A), glycine (G), and serine (S) as very small; cysteine (C), proline (P), aspartate (D), asparagine (N), and threonine (T) as small; valine (V), glutamate (E), histidine (H), and glutamine (Q) as medium; isoleucine (I), leucine (L), methionine (M), lysine (K), and arginine (R) as large; and phenylalanine (F), tryptophan (W), and tyrosine (Y) as very large [46]. Furthermore, based on the wild-type and mutated amino acid classes, a unique label is calculated as AASCI(wild-type) × 5 + AASCI(mutant), where AASCI(x) gives the amino acid size class index of amino acid x. Thus, this feature uses 25 (5 × 5) labels covering all possible 20 × 20 combinations of wild-type and mutant amino acids.

Mutation hydrogen-bonding class. This is a categorical feature for which the 20 standard amino acids are grouped into four classes: isoleucine (I), leucine (L), methionine (M), valine (V), cysteine (C), proline (P), phenylalanine (F), alanine (A), and glycine (G) as non-hydrogen-bonding; lysine (K), arginine (R), and tryptophan (W) as hydrogen-bond donors; histidine (H), glutamine (Q), asparagine (N), threonine (T), tyrosine (Y), and serine (S) as hydrogen-bond donor-acceptors; and glutamate (E) and aspartate (D) as hydrogen-bond acceptors [46]. Furthermore, based on the wild-type and mutated amino acid classes, a unique label is calculated as HBCI(wild-type) × 4 + HBCI(mutant), where HBCI(X) gives the hydrogen-bonding class index of amino acid x. Thus, this feature uses a total of 16 (4 × 4) labels covering all possible 20 × 20 combinations of wild-type and mutant amino acids.

Mutation chemical-type class. This is a categorical feature for which the 20 standard amino acids are grouped into seven classes based on the chemical type of the side chain: lysine (K), arginine (R), and histidine (H) as basic; glutamine (Q) and asparagine (N) as amide; aspartate (D) and glutamate (E) as acidic; methionine (M) and cysteine (C) as sulfur-containing; serine (S) and threonine (T) as hydroxyl; tryptophan (W), tyrosine (Y), and phenylalanine (F) as aromatic; and isoleucine (I), leucine (L), valine (V), proline (P), alanine (A), and glycine (G) as aliphatic. Furthermore, based on the wild-type and mutated amino acid classes, a unique label is calculated as CTCI(wild-type) × 7 + CTCI(mutant), where CTCI(x) gives the chemical-type class index of amino acid x. Thus, this feature uses 49 (7 × 7) labels covering all possible 20 × 20 combinations of wild-type and mutant amino acids.

Mutation type class: This is a categorical feature that uses 400 different labels for all possible combinations of 20 standard wild-type and mutant amino acids.

(IV)Accessibility of the mutation site

The accessibility of the mutation site from the wild-type protein–DNA complex structure was calculated using mkdssp (v2.0.4) [47,48].

(V)Accessibility changes due to mutation

The accessibility of the mutation site for the wild type (acc_wild-type_) and mutant (acc_mutant_) was calculated from the wild-type protein–DNA complex structure and modeled mutated protein–DNA complex structure, respectively, using mkdssp (v2.0.4). Then, the accessibility change (delta_acc) was calculated as acc_wild-type_ − acc_mutant_.

(VI)Backbone torsion angles of the mutation site

Two features corresponding to backbone torsions (Φ and Ψ) of the mutation site of the protein–DNA complex structure were calculated using mkdssp (v2.0.4).

(VII)Protein secondary structure composition

The seven-class secondary structure for each protein residue in the protein–DNA complex was predicted using mkdssp (v2.0.4). The ratio of counts of amino acids adopting a given secondary structure to the total number of residues in the protein structure was then calculated. In our dataset, the ratio for pi-helix was zero in all mutation cases. Thus, it was discarded, leaving a list of ratios of six secondary structures: α-helix, isolated β-bridge, extended bridge participating in the β-ladder, 310-helix, hydrogen-bonded turn, and bend.

(VIII)Protein–DNA contact features

The protein–DNA wild-type complex structure was analyzed using the x3dna-dssr (v2.4.5) snap program [49], and the following four total interactions were extracted from the output and used as four features.

  (a)Nucleotide amino acid contacts: The total number of interactions between the protein and the DNA.  (b)Base amino acid hydrogen bonds: The total number of hydrogen-bond nucleotide bases in the DNA and protein residues.  (c)Phosphate amino acid hydrogen bonds: The total number of hydrogen bonds between the nucleotide phosphate and protein residues.  (d)Base amino acid stacks: The total number of stacks identified in the protein–DNA complex structure between the nucleotide bases and protein residues.

(IX)Changes in protein–DNA contacts due to a mutation

To calculate these features, we used the mutated protein–DNA complex structure with a single amino acid mutation in the protein. First, a mutated protein structure was modeled using the wild-type protein structure extracted from the complex. We used the Scwrl4 (v4.0.2) [50] side-chain modeling tool for this purpose. Scwrl4 (v4.0.2) requires the input protein structure to have only standard amino acids with a complete backbone for each residue. To meet these requirements, the extracted protein structure was cleaned by purging residues with incomplete backbones, followed by renaming the non-standard amino acids to their parent standard amino acids and listing the corresponding atoms with the “ATOM” record if they were listed as “HETATM” in the extracted protein structure. Afterward, the side chains of non-standard amino acids were reverted to standard, using Scwrl4 (v4.0.2), to yield a cleaned wild-type protein structure. Second, the cleaned wild-type protein structure was superposed to the input protein–DNA complex over the common backbone atoms in the proteins of both structures. All the non-protein atoms were copied to the cleaned wild-type protein structure to produce a cleaned wild-type protein–DNA complex. Similarly, the single amino acid mutant protein structures were modeled using Scwrl4 (v4.0.2), and the complex structure was compiled after superposition, as described earlier.

The protein–DNA wild-type complex structure and mutated protein–DNA model structures were analyzed using the x3dna-dssr (v2.4.5) snap program, and four total interactions were extracted from the output for both wild-type and mutation cases. The differences between the wild-type and mutant were used as four delta features.

  (a)Delta nucleotide amino acid contacts: The total change in the number of interactions between the protein and the DNA due to a mutation.  (b)Delta base amino acid hydrogen bonds: The total change in the number of hydrogen-bond nucleotide bases in the DNA and protein residues due to a mutation.  (c)Delta phosphate amino acid hydrogen bonds: The total change in the number of hydrogen bonds between the nucleotide phosphate and protein residues due to a mutation.  (d)Delta base amino acid stacks: The total change in number of stacks identified in the protein–DNA complex structure between the nucleotide bases and protein residues due to a mutation.

This resulted in 49 features, subsequently used for training the machine learning tools to elucidate the effects of the protein mutation on ΔΔG. The dataset S596, which consists of 596 data points, was used for training and 5-fold cross-validation.

#### 2.4.2. DNA Mutation

The features associated with protein–DNA interactions were grouped according to their characteristics. These features were used to develop the model for predicting changes in the binding free energy due to the mutation in the DNA base pair in the protein–DNA complex.

(I)Protein structure features

The protein secondary structure ratios were defined and calculated as described in Section 2.4.1—Point (VII) and were also used for predicting changes in the binding free energy due to DNA base-pair mutations.

(II)DNA structural feature of the mutation site

There are 18 features related to the DNA base-pair structure at the mutation site, including six base-pairing parameters (shear, stretch, stagger, buckle, propeller, and opening), six base-pair step parameters (shift, slide, rise, tilt, roll, and twist), and six base-pair helicity parameters (x-displacement, y-displacement, helical rise, inclination, tip, and helical twist), which were calculated using x3dna-dssr (v2.4.5) for the wild-type protein–DNA complex structure. Variations in these parameters impact the base-pairing strength at the DNA mutation site, affecting the protein–DNA binding strength. However, these parameters are defined for only double-stranded DNA. For protein–DNA complexes consisting of single-stranded DNA, these 18 features are assigned a value of zero.

(III)DNA mutation categorical features
(a)Base-pair type: The base pairs are grouped into two classes, with AT and TA into pairs bonded with two hydrogen bonds and GC and CG bonded with three hydrogen bonds. Two labels are given a value of zero for AT or TA and one for a GC or CG pair. This feature encodes a wild-type base pair.(b)Wild or mutant base-pair mismatch: This feature encodes the matched/mismatched base-pair status of wild and mutant base pairs. If both the wild-type base pair and mutant base pair match (i.e., sets AT, TA, CG, or GC), the label “zero” is used; otherwise, the label “one” is used.(c)Mutant base-pair category: This categorical feature uses 256 (i.e., 16 × 16) different labels to encode the wild-type to mutant base pairs. There are 16 possible base pairs of four nucleotides. These 16 base pairs are assigned 16 distinct indices from 0 to 15. The wild-type base-pair/mutant base-pair label index is calculated as BPI(wild-type base-pair) × 16 + BPI(mutant base-pair), where BPI(XY) represents the base-pair index of XY.(IV)Protein–DNA interaction features

We also used the four features capturing the number of protein–DNA contacts and hydrogen bonds, as defined and calculated in Section 2.4.1—Point (VIII) for the wild-type protein–DNA structure. These features account for contacts between any DNA base pair and any protein residue in the protein–DNA complex structure.

(V)Protein mutation site forward strand base interaction features

These four features considered only the contacts and hydrogen bond features defined in Section 2.4.1—Point (VIII), but involving only the forward strand DNA base at the mutation site and any protein residue in the protein–DNA complex structure.

This resulted in 35 features, which were used for training the machine learning tools to elucidate the effects of the DNA mutation on ΔΔG. The D502 dataset, which consists of 502 data points, was used for this purpose.

Appendix A provide a categorization of features used to predict the changes in the binding free energy due to mutations of a single amino acid in the protein and single base/base pair in the DNA, respectively.

### 2.5. Machine Learning Model Training

The complete set of features described in Section 2.4, along with the target variable (the binding free energy change, ΔΔG), forms the dataset for the machine learning training, testing, and model development. Two distinct datasets, comprising mutations in protein and DNA within the complexes and referred to as S596 and D502, respectively, were trained independently. This approach is expected to result in the development of two separate models tailored to these specific cases.

A critical objective in developing any machine learning model is to avoid overfitting or underfitting. To achieve this, the dataset is partitioned into multiple folds, using a technique known as *k*-fold cross-validation. In this method, the dataset is divided into *k* equal-sized subsets (or folds), where *k*-1 folds are used for training, and the remaining fold is reserved for cross-validation. The process is iterated *k* times, ensuring that each fold is used for training and testing. Model performance is evaluated during each iteration, using metrics such as Pearson correlation coefficient (PCC) and root mean square error (RMSE). This systematic approach ensures robust model evaluation and minimizes the risk of overfitting, contributing to reliable predictive performance.

### 2.6. Selecting the Optimal Machine Learning Approach Using PyCaret

Predicting a continuous numerical variable—the change in binding free energy (ΔΔG)—necessitates using regression-based machine learning methods. To select the best regression method, we used the AutoML functionality of the PyCaret [51] library, which facilitates a quick and preliminary assessment of the performance of various regression algorithms on the training dataset—a task which was not undertaken in the previous version of SAMPDI-3D. This approach ensures a streamlined and efficient evaluation process by leveraging default hyperparameters and an automated training pipeline. The protein and DNA mutation datasets, along with their respective feature sets, were trained using 5-fold cross-validation. Table 1 and Table 2 summarize the performance metrics for the top 10 regression algorithms for the protein and DNA mutation datasets, ranked by their predictive accuracy.

### 2.7. Hyperparameter Tuning and Advanced Model Training

Following the preliminary results from AutoML, the top seven models, ranked based on the PCC between the experimental and predicted ΔΔG values, were selected for further training with extensive hyperparameter tuning. Hyperparameters (i.e., pre-determined parameters governing how a model learns and operates during training) are critical in preventing overfitting or underfitting. A comprehensive dictionary of potential hyperparameter values was constructed to optimize the models. Each model underwent rigorous training with 1000 iterations of 5-fold cross-validation. During each iteration, a unique combination of hyperparameters was randomly sampled from the dictionary, enabling the identification of the optimal hyperparameter set for which the maximum PCC between the actual and predicted values was obtained.

Table 3 and Table 4 summarize the performance metrics for the best-performing iterations of all seven algorithms, applied to the protein and DNA mutation datasets. Extreme gradient boosting (XGBoost) achieved PCCs of 0.67 and 0.78 for the protein and DNA mutation datasets, respectively. The hyperparameters corresponding to these best-performing iterations will be used in subsequent studies.

Based on these results, extreme gradient boosting (XGBoost) [52] was selected as the regressor model for further analysis because of its superior performance in predicting ΔΔG values across both datasets. XGBoost is an advanced implementation of gradient-boosted decision trees. It is known for its efficiency, scalability, and ability to balance computational speed with high predictive accuracy, making it ideal for handling large and complex datasets.

## 3. Results

### 3.1. Dataset (Protein Mutations) Comparison of SAMPDI-3D and the Newly Curated Dataset from ProNAB

S419 is the protein mutation dataset obtained from our SAMPDI-3D training dataset, whereas S177 is a newly curated protein mutation dataset compiled from ProNAB. We followed the staged filtering method while curating the S177 dataset mentioned in the data cleaning section. We compared the mutated residue counts and percentages (Table 5) between our SAMPDI-3D training dataset (S419) and the newly created S177 dataset.

In previous works, we noted that various methods for estimating changes in the binding free energy due to mutations have varied levels of sensitivity to different types of mutations [53,54]. This variability arises from the unequal representation of different amino acids in the training datasets. Hence, we compared the mutated amino acid residues between the S419 and S177 datasets and identified clear differences in the types and frequencies of amino acid mutations (Table 5). This outcome motivated us to carefully design features for machine learning model development. Alanine (A) mutations were common in both datasets. However, S419 had a higher frequency (70.64%) than S177 (61.02%). Cysteine (C) mutations were present in S419 (1.43%) but not in S177. S177 had a higher frequency of aspartic acid (D) and glutamic acid (E) mutations, with the latter accounting for 7.34% vs. 1.43% for aspartic acid in S419.

Methionine (M) mutations occurred more frequently in S177 (3.95%) than in S419 (1.43%). Glycine (G) and proline (P) mutations were more common in S177, whereas histidine (H) mutations were more common in S419 (1.43%) than in S177 (0.56%). S419 contained mutations in isoleucine (I), tryptophan (W), and tyrosine (Y), but S177 did not (Table 5).

Other amino acids, including phenylalanine (F), lysine (K), leucine (L), asparagine (N), glutamine (Q), arginine (R), serine (S), threonine (T), and valine (V), exhibited similar frequencies in the two datasets, with some slight changes. Overall, S419 had a higher percentage of alanine and histidine mutations, whereas S177 showed greater diversity, with higher frequencies of glutamic acid, glycine, methionine, and other alterations, illustrating the differences between the mutation profiles of the datasets.

For a consolidated quantitative comparison of the diversity (representativeness) of the two datasets (S419 and S177), we computed the Shannon entropy based on the probability of each of the 20 mutating amino acids. Shannon entropy is calculated as $H={-\Sigma }_{a\;\in \;amino\;acids}\;p_{a}ln(p_{a})$, where $p_{a}$ is the probability of a particular amino acid a. A higher Shannon entropy implies greater diversity or closeness to the distribution being equiprobable as information is more evenly distributed across categories. The Shannon entropy for S419 was 1.432 nats, while for S177, it was 1.663 nats, indicating greater diversity in S177. To provide context for these values, the maximum entropy—achieved when all 20 amino acids are equally represented in the dataset—was 2.996 nats.

Overall, the S419 dataset, taken from our SAMPDI-3D training data, exhibited a high proportion of alanine mutations (70.64%) and overall lower diversity or representativeness, indicating a different mutation profile. By contrast, the recently curated S177 dataset from ProNAB exhibited increased diversity, including numerous glutamic acid, glycine, and methionine variants, providing a larger mutation spectrum for investigation.

### 3.2. Performance of SAMPDI-3D and Other Available Methods Tested on the S177 and D42 (Newly Curated from ProNAB) Datasets

The S177 dataset consisted of 177 new single amino acid mutations in proteins, and D42 consisted of 42 new single-base/base-pair mutations in the DNA of the protein–DNA complex, obtained by cleaning ProNAB. We first assessed the performance of the state-of-the-art methods for single-mutation-induced changes in the binding free energy of the protein–DNA complex, using these two new datasets to learn where they stand and identify points for improvement.

Using SAMPDI-3D to predict changes in the binding free energy due to point mutations in the protein involved in protein–DNA binding, we obtained a PCC of 0.17 and RMSE of 1.34 kcal/mol. Similarly, mCSM-NA and PremPDI resulted in a PCC of 0.34 and RMSE of 1.31 kcal/mol and a PCC of 0.36 and RMSE of 1.35 kcal/mol, respectively (Figure 2 and Table 6). In the case of a single-base/base-pair mutation in DNA, a PCC of 0.71 was obtained using SAMPDI-3D (Figure 3 and Table 6). However, among the listed methods—apart from SAMPDI-3D—only PNBACE predicted the changes in binding free energy due to mutations in the DNA of a protein–DNA complex. However, the extended computational time (up to 24 h for a single mutation) of the PNBACE web server prevented predicting ΔΔG within a reasonable timeframe when using the D42 dataset. This limited us to using only SAMPDI-3D for predicting ΔΔG for DNA mutations listed in the D42 dataset (Figure 3 and Table 6). Note that SAMPDI-3D and PremPDI define the binding free energy change as ΔΔG = ΔG(mutant) − ΔG(wild-type). By contrast, mCSM-NA defines ΔΔG = ΔG(wild-type) − ΔG(mutant). Thus, we reversed the sign of the predicted ΔΔG values obtained from mCSM-NA to conduct the analysis to ensure a consistent comparison with other prediction methods.

SAMPDI-3D was trained on S419 for the protein mutation dataset and D463 for the DNA mutation dataset and achieved PCCs of 0.76 (MSE 0.53 kcal/mol) and 0.80 (MSE 0.39) for the protein and DNA mutations [29], respectively. Such results are expected, considering that the S419 dataset is less diverse than the new S177 (see Section 3.1). However, although the performance of the D42 dataset is not as good as that of D463, it is still comparable, suggesting there is a scope for further extending the feature set to cover underrepresented categories.

The performance assessment of the existing methods, including SAMPDI-3D, using the newer experimental data sets indicated that they need to improve their performance. This motivated us to train the SAMPDI-3D further on expanded datasets and use an extended feature set to advance its prediction capability.

### 3.3. Performance of SAMPDI-3D v2 Tested on Protein and DNA Mutation Databases

A total of 50 iterations of 5-fold cross-validation were conducted separately for the protein and DNA mutation datasets, using the optimized hyperparameters obtained during the tuning phase. For this purpose, 49 features were used for the protein dataset and 35 for the DNA mutation dataset.

For each iteration of the 5-fold cross-validation (k = 5), separate models were trained by designating one of the five folds as the validation set and using the remaining folds for training. Each model generated predictions for the cross-validation set during the iteration, which were compiled into a list. This approach ensured that each data point appeared in the validation set exactly once and was excluded from training while predicting data points from the associated fold during that iteration. Performance metrics, like the PCC, RMSE, and the slope and intercept of the fitted line, were calculated based on the combined predictions set.

For the best-performing iteration, PCCs of 0.68 and 0.80 were achieved for the protein and DNA mutation datasets, respectively (the metrics were calculated as detailed in Section 2.5). The final prediction model was constructed using an ensemble prediction approach, obtained by averaging the models’ predictions across all five folds of the best-performing iteration. The consensus model was implemented in both the web server and standalone code. Table 7 summarizes the performance metrics of the best-performing iteration and the average performance across all 50 iterations. Furthermore, plots were generated for the best-performing iterations for the whole datasets in both cases (Figure 4).

### 3.4. Web Server Implementation

The updated version of SAMPDI-3D (SAMPDI-3D v2) is freely available at http://compbio.clemson.edu/SAMPDI-3Dv2/ (accessed on 26 December 2024). It features an easily accessible, user-friendly interface for job submissions. Users can upload the PDB structure of a complex and provide the chains involved in the complex, extracted from the input PDB, using pdb-tools [55]. Then, the user specifies a single mutation or a batch of mutations to obtain the predicted ΔΔG values directly from the website. A sample format for specifying mutations is also provided on the website. Additionally, a stand-alone code is provided for download, allowing the users to make local predictions and modifications according to their goals.

## 4. Discussion

The work led to a new, more accurate version of the SAMPDI-3D method. The enhanced version has additional features that improve the specificity of predictions, particularly for DNA mutations, and is trained on a larger database. In the new development, users are given additional information as the 3D structure of the mutant protein. Furthermore, the final prediction model in the new SAMPDI-3D was constructed using an ensemble prediction approach, obtained by averaging the predictions of models across all five folds of the best-performing iteration, in contrast to the old SAMPDI-3D, where the best model was implemented. The algorithm is available as a web server and as a stand-alone code. The computational time for delivering a prediction is only a few minutes per complex, making SAMPDI-3D v2 a tool that can be applied for genome-scale investigations.

Database cleaning is an important component of the present work. While it is tempting to use a large database consisting of entries automatically collected from the literature, such a database contains many wrong entries—either wrong ΔΔGs or wrong position and type of mutation, just to mention some examples. Thus, directly using the uncleaned database for training new algorithms comes with a greater risk of inaccurate predictions because of accumulated noise from inaccurate data points. In turn, using these predicted ΔΔGs may lead to the wrong conclusions while assessing the impact of mutations and their pathogenicity.

## Figures and Tables

**Figure 1 genes-16-00101-f001:**
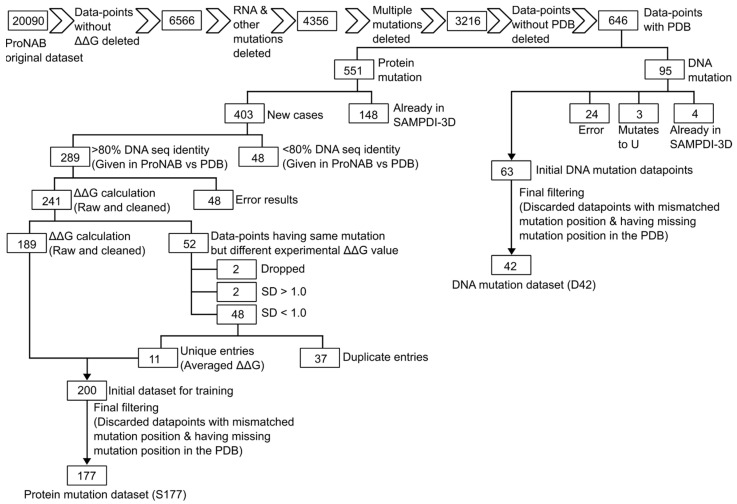
Flowchart of the ProNAB database cleaning to remove entries with insufficient information, duplicated entries, and typos.

**Figure 2 genes-16-00101-f002:**
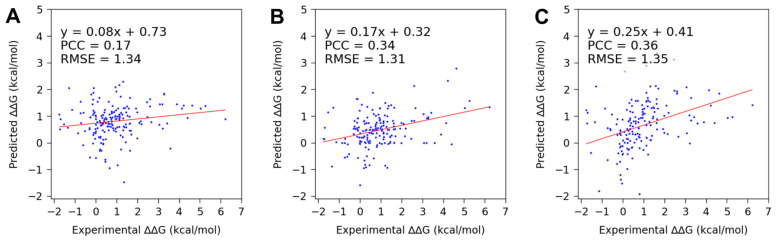
Scatter plots showing the trend lines and fitted parameter values of the actual and predicted changes in binding free energy for the newly curated protein mutation dataset (S177) from ProNAB, using (**A**) SAMPDI-3D, (**B**) mCSM-NA, and (**C**) PremPDI. The data points are the same for SAMPDI-3D and mCSM-NA. However, in the case of PremPDI, the prediction failed for 13 data points, leaving only 164 to analyze.

**Figure 3 genes-16-00101-f003:**
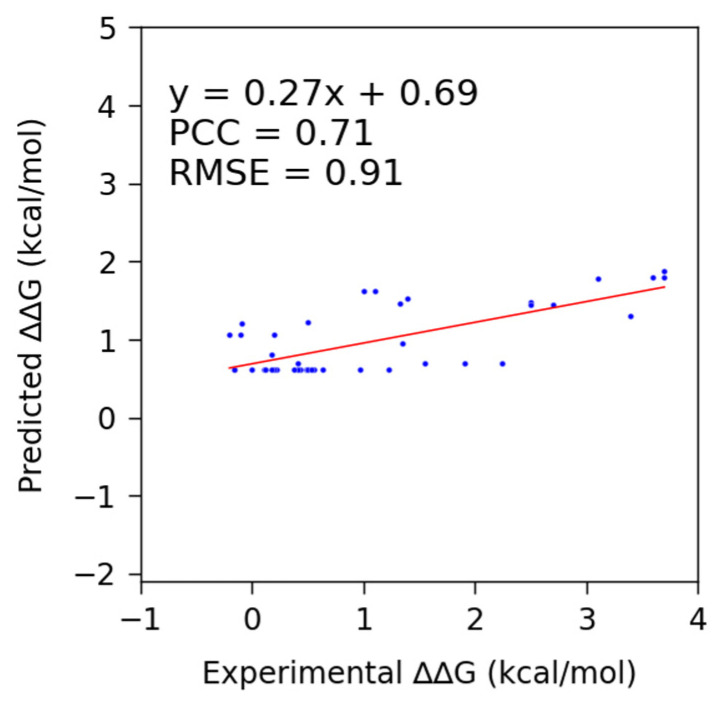
Scatter plots showing the trend lines and fitted parameter values of the actual and predicted changes in binding free energy, using SAMPDI-3D for the newly curated DNA mutation dataset (D42) from ProNAB.

**Figure 4 genes-16-00101-f004:**
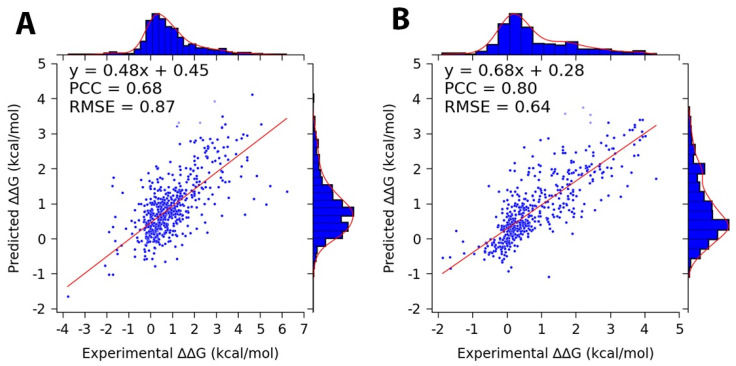
Scatter plots showing the marginal distributions and trend lines with fitted parameter values for the actual and predicted changes in binding free energy, obtained from the best-performing iterations of the (**A**) protein (S596) and (**B**) DNA (D502) mutation datasets.

**Table 1 genes-16-00101-t001:** The top 10 regression algorithms from PyCaret trained on the protein mutation database based on performance (PCC) in the screening phase.

Model	PCC	RMSE (kcal/mol)
CatBoost regressor	0.65	1.32
Extra trees regressor	0.65	1.28
Gradient boosting regressor	0.64	1.16
Light gradient boosting machine	0.64	1.24
Random forest regressor	0.63	1.28
Extreme gradient boosting	0.63	1.28
AdaBoost regressor	0.61	1.30
Linear regression	0.46	1.46
Ridge regression	0.43	1.49
Huber regressor	0.32	0.56

**Table 2 genes-16-00101-t002:** The top 10 regression algorithms from PyCaret trained on the DNA mutation database based on performance (PCC) in the screening phase.

Model	PCC	RMSE (kcal/mol)
CatBoost regressor	0.71	0.75
Random forest regressor	0.69	0.77
Extra trees regressor	0.68	0.79
Gradient boosting regressor	0.67	0.79
Extreme gradient boosting	0.67	0.79
Light gradient boosting machine	0.65	0.81
K neighbors regressor	0.62	0.84
AdaBoost regressor	0.59	0.86
Decision tree regressor	0.45	0.94
Decision tree regressor	0.45	0.98

**Table 3 genes-16-00101-t003:** Top seven regression algorithms trained on the protein mutation database, ranked by their performance (PCC), after thorough hyperparameter tuning.

Model	PCC (Best Iteration)	RMSE (kcal/mol) (Best Iteration)
Extreme gradient boosting	0.67	0.89
CatBoost regressor	0.66	0.91
Gradient boosting regressor	0.66	0.89
Extra trees regressor	0.65	0.91
Light gradient boosting machine	0.65	0.9
AdaBoost regressor	0.64	0.92
Random forest regressor	0.63	0.91

**Table 4 genes-16-00101-t004:** Top seven regression algorithms trained on the DNA mutation database, ranked by their performance (PCC), after thorough hyperparameter tuning.

Models	PCC (Best Iteration)	RMSE (kcal/mol) (Best Iteration)
Extreme gradient boosting	0.78	0.69
CatBoost regressor	0.77	0.69
Light gradient boosting machine	0.77	0.7
Random forest regressor	0.76	0.7
Extra trees regressor	0.76	0.7
Gradient boosting regressor	0.76	0.7
K neighbors regressor	0.74	0.73

**Table 5 genes-16-00101-t005:** Percentages and count table of the mutated residue in the S419 and S177 datasets.

Mutated Residue	S419 (Count)	S419 (%)	S177 (Count)	S177 (%)
Alanine (A)	296	70.64	108	61.02
Cysteine (C)	6	1.43	0	0
Aspartic acid (D)	5	1.19	4	2.26
Glutamic acid (E)	6	1.43	13	7.34
Phenylalanine(F)	9	2.15	4	2.26
Glycine (G)	9	2.15	7	3.95
Histidine (H)	6	1.43	1	0.56
Isoleucine (I)	2	0.48	0	0
Lysine (K)	13	3.1	5	2.82
Leucine (L)	12	2.86	4	2.26
Methionine (M)	6	1.43	7	3.95
Asparagine (N)	5	1.19	3	1.69
Proline (P)	2	0.48	2	1.13
Glutamine (Q)	7	1.67	5	2.82
Arginine (R)	9	2.15	3	1.69
Serine (S)	9	2.15	5	2.82
Threonine (T)	6	1.43	3	1.69
Valine (V)	7	1.67	3	1.69
Tryptophan (W)	1	0.24	0	0
Tyrosine (Y)	3	0.72	0	0

**Table 6 genes-16-00101-t006:** Performance of SAMPDI-3D and other methods tested on newly curated data points from ProNAB in the S177 (protein) and D42 (DNA) datasets.

Mutation	Method	PCC	RMSE
Protein	SAMPDI-3D	0.17	1.34
	mCSM-NA	0.34	1.31
	PremPDI	0.36	1.35
DNA	SAMPDI-3D	0.71	0.91

**Table 7 genes-16-00101-t007:** Performance metrics for the best-performing iteration and average metrics for 50 iterations of 5-fold cross-validation for the protein and DNA mutation datasets.

Mutation	PCC (Best Iteration)	Average PCC (50 Iterations)	Number of Features
Protein	0.68	0.65 ± 0.05	49
DNA	0.80	0.77 ± 0.06	35

## Data Availability

The S177 and D42 datasets can be downloaded from http://compbio.clemson.edu/SAMPDI-3Dv2/, accessed on 26 December 2024.

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
