# Peer review of "Further Development of SAMPDI-3D: A Machine Learning Method for Predicting Binding Free Energy Changes Caused by Mutations in Either Protein or DNA"

_genes, 2025, doi:10.3390/genes16010101_

Round 1

Reviewer 1 Report

Comments and Suggestions for Authors

This is a timely contribution which addresses an important topic of predicting the effect of mutations on protein-DNA binding free energy. It has direct impact on understanding molecular effects of disease-causing mutations and on protein-DNA engineering. The methodology is clearly outlined and careful benchmarking is carried out to assess the performance of the updated SAMPDI-3D. The paper makes good impression by providing compelling evidence of necessity of SAMPDI-3D development by showing that neither of existing methods, including the old SAMPDI-3D, can predict the change of the binding free energy on the new entries in ProNAB database. Teh senior author, Alexov, is a top expert in the field. 

 Major comments:

None

Minor comments:

1)      While the manuscript clearly points out the limitations of ML methods trained on small number of cases, the statement could be strengthened by referring to https://pubmed.ncbi.nlm.nih.gov/37384816/

2)      Lines 262-263. There is a missing reference in the sentence “….the D463 dataset (ref our paper)….”. Must be corrected.

3)      Line 301. “(II) Mutation type related Features”. Features should not be capitalized.

4)      The words in subsections are sometime capitalized or not capitalized. This should be made consistent according to the journal style.

5) in Figures 2-5, please make the labels in the x-axes upright. 

Author Response

We thank the reviewer for useful comments and suggestions. They are all taken into account and appropriate changes are made. Below we address the reviewer's comments point-by-point. 

1)      While the manuscript clearly points out the limitations of ML methods trained on small number of cases, the statement could be strengthened by referring to https://pubmed.ncbi.nlm.nih.gov/37384816/

Ans: Thank you for your comment. The reference has been included in the paper in lines 148-150.

2)      Lines 262-263. There is a missing reference in the sentence “…. the D463 dataset (ref our paper)….”. Must be corrected.

Ans: Thank you for your comment. The reference has been included in the paper in line 257.

3)      Line 301. “(II) Mutation type related Features”. Features should not be capitalized.

Ans: The capitalization has been removed.

4)      The words in subsections are sometime capitalized or not capitalized. This should be made consistent according to the journal style.

Ans: According to the journal requirement, capitalization has been removed from the title of the subsection.

5) in Figures 2-5, please make the labels in the x-axes upright.

Ans: The X-axis labels have been made upright.

Reviewer 2 Report

Comments and Suggestions for Authors

This paper represents a significant improvement in the ability to predict ΔΔG values for protein/DNA binding for single mutations in either protein or DNA. Independently of the improvements in the SAMPDI-3D method, this paper also presents valuable statistical analyses of mutations impacting DNA/protein binding as well as a useful example of how to clean databases for machine learning.

SAMPDI-3D v2 improves upon the authors' previous method by incorporating newly available data in training their method. However, there are a few uncertainties in how this method is presented that need to be resolved prior to publication. Furthermore, the authors need to correct usage and other errors before this paper is ready for publication.

Line 53 and following occurrences: Kd is usually stylized as KD or Kd.

In lines 62, 73 & 74, the authors use the phrase, “large amount of samples”. I am curious as to how much exactly is a large amount (e.g. in the context of DNA/protein binding studies).

In line 76, the authors need to indicate what TF is an abbreviation for (and see section on English language revisions for more about this sentence).

Lines 157-159: It sounds like what the authors did was to retrain the SAMPDI-3D on the latest database (and then to test it thoroughly). Indeed, by line 162, it is clear the authors did add new features. Perhaps the authors could revise the last two paragraphs of the introductiojn (maybe merging them into one) in such a way as to highlight all of the novel aspects of this latest version of SAMPDI-3D while still maintaining some sense that the new features were added based on the results with the retrained program.

Lines 292-298: If these calculations are not described in reference 40, the reference given for Blast, the authors should provide a reference (e.g. in either line 292 or at the use of the term PSSM matrix in line 289) to the source from which they learned (or could have learned) how to perform these calculations.

Line 304: Why did the authors choose Moon’s hydrophobicity index? Is this index the one typically used in analyzing protein/DNA interactions or is there another reason. I am not sure if it is necessary to include a parenthetical one phrase explanation of why they chose this index, but I suspect it some readers may wish to know the reason.

Unless I am missing something, the total number of features used to predict ΔΔG values is not explicitly stated until line 618. It should be stated somewhere in Section 2.4. I imagine that the machine learning techniques used in this paper are not as sensitive to the need for the number of datapoints to greatly exceed the number of independent variables as traditional regression techniques fitting one parameter per variable (possibly with an additional offset parameter as well as additional interaction parameters), and the cross-validation would detect if overfitting did occur as a result of insufficient degrees of freedom. However, I would imagine the feature vector still should have a lower dimensionality than the number of points in the training data set. Is this the case?

Section 2.6 leaves the reader a bit unclear. Are the authors using a different approach than they did for the original version of SAMPDI-3D? If the approach is the same, it may be possible to abbreviate this section. If the approach is different, it should be noted. Is the code written so that users can customize which machine learning approach is used? Similar questions can be raised about Section 2.7, but so long as the answers are the same as for Section 2.6, there need not be any changes to Section 2.7.

I thank the authors for providing the interesting analysis in Section 3.1. Has any similar analysis been performed previously? If so, the authors might want to reference it and compare their training set with previously analyzed datasets. If not, I do not think any changes are needed to this section.

Do the results presented in Section 3.2 indicate that the original SAMPDI-3D method was overfit on the S419 dataset? How did the other methods used in this section perform on that dataset? Relatedly, how did the other methods perform on the S596 dataset? The abstract of the paper claims that SAMPDI-3D v2 performs significantly better than other methods, but the only PCC values I see for other methods are obtained on the S177 dataset (Table 6). If the authors wish to claim significance, they should provide the PCC for the best performing other predictor (just as a number, preferably with some indication of its uncertainty) on the S956 dataset and compare it with the 0.65 ± 0.05 figure obtained from SAMPDI-3D v2. At the very least, the authors should compare the SAMPDI-3D v2 average correlation coefficient of 0.65 with the correlation coefficient for the best performing model on S596 using (although it is not perhaps a statistically rigorous use of this test) using Fisher r to z transformations. For what it’s worth, assuming that PremPDI remains the best performing model and the correlation coefficient on the S596 data for that method remains around 0.4, on a dataset with 596 datapoints, the z score for comparing a correlation coefficient of 0.4 with a correlation coefficient of 0.65 is > 6, which is highly significant by any standard.

In the Discussion section, the authors point out the importance of using accurate ΔΔG values to train their method and emphasize the importance of their database cleaning. However, it could be the case that, despite their careful cleaning, some of the ΔΔG values they used were inaccurate. Indeed, in the case of mutations with multiple associated ΔΔG values, it could be a rejected value that is correct. While the cross-validation performed would tend to support the overall accuracy of the model, I am curious as to whether the authors explored any of the cases where, during cross-validation, the model failed to accurately predict ΔΔG. Are there any cases in which cross-validation uncovered an error in the dataset used in this paper? In a similar vein, did the authors apply their method to rejected ΔΔG values to verify that they are indeed not what would be predicted from accepted ΔΔG values?

I am also curious (although it may be beyond the scope of this paper, and there may not be enough data to support this analysis) as to whether the success in prediction of ΔΔG values depends on the experiment used to obtain those ΔΔG values. Under the assumption that the predictions are overall accurate, do some experimental methods appear to give more accurate ΔΔG values than others?

Comments on the Quality of English Language

In line 75, there should be a new paragraph beginning with “In addition to traditional methods” as the paragraph, as it currently stands, is very long.

Lines 76ff: “For example, protein binding microarrays (PBMs) is a fluorescence-based methods that assess the binding specificities of TFs to interacting DNA”. I am not sure if PBMs is a particular method or not, which impacts whether the verb form used should be “is” or “are”. I suspect “methods” needs to be singular (“method”) and perhaps the sentence should just be rephrased to sound less awkward no matter what the correct number agreement is.

Line 86: “it’s” should be “its”.

The paragraph beginning on line 101 is also very long and perhaps should be split up. The obvious place (to me, at least) to split is it to begin a new paragraph with “Other models available” (splitting the authors’ approaches from other approaches) on line 111. This splits the paragraph a wee bit more asymmetrically than would be desirable, however.

Line 145f: “no other methods can predict the effects of mutation in DNA on ΔΔG” is a bit strong. Perhaps “none of the other methods mentioned above can predict the effects of DNA mutation on ΔΔG” would be better?

Line 161: Should be “Results section” (not sure if section should be capitalized or not) as opposed to “result section”.

Lines 168-170: “Since our model development requires experimentally measured changes of binding free energy (ΔΔG) upon mutation of either in protein or DNA” is a sentence fragment.

Line 227f: the phrase “Additionally, 34 data points had issues with the PDB components (e.g., missing residues), and they were removed,” can be shortened to “Additionally, we removed 34 data points that had issues with the PDB components (e.g., missing residues),”. Another possible change (to eliminate a couple of words) is found on line 240f: “In our previous version of SAMPDI-3D, the S419 dataset was used for training and testing of ΔΔG caused by protein mutation.”, which can be edited to “Our previous version of SAMPDI-3D used the S419 dataset for training and testing of ΔΔG caused by protein mutation.” The authors should double check that they are as succinct as possible to keep the paper a manageable size should they have to add additional material in response to reviewers.

Line 261f: I see that I am not the only one who uses parentheticals such as “(ref our paper)” in the drafting process and then forgets to actually replace them with the actual reference in the submitted manuscript. Of course, this should be fixed during the revision.

Lines 652ff: The sentence “The work resulted in a new version of the SAMPDI-3D method, which is now more accurate than the previous version, equipped with new features to strengthen the specificity of the predictions, especially for mutations in the DNA and trained on a larger database.” is long and awkward, and could use revision.

Author Response

We thank the reviewer for useful comments and suggestions. They are all taken into account and appropriate changes were made. Below we address reviewer's comments point-by-point. 

  1. Line 53 and following occurrences: Kd is usually stylized as KD or Kd.

Ans: The style for Kd has been updated to Kd throughout.

  1. In lines 62, 73 & 74, the authors use the phrase, “large amount of samples”. I am curious as to how much exactly is a large amount (e.g. in the context of DNA/protein binding studies).

Ans: We have updated the text to add the amount of samples needed for experiments along with the reference in line 71-73.

  1. In line 76, the authors need to indicate what TF is an abbreviation for (and see section on English language revisions for more about this sentence).

Ans: The abbreviation TF was introduced in line 35 and stands for Transcription Factor. English has been improved as well.

  1. Lines 157-159: It sounds like what the authors did was to retrain the SAMPDI-3D on the latest database (and then to test it thoroughly). Indeed, by line 162, it is clear the authors did add new features. Perhaps the authors could revise the last two paragraphs of the introduction (maybe merging them into one) in such a way as to highlight all of the novel aspects of this latest version of SAMPDI-3D while still maintaining some sense that the new features were added based on the results with the retrained program.

Ans: As suggested by the reviewer, we combined the last paragraphs of the introduction section to indicate the necessity of SAMPDI-3D further development, and the additional steps made.

  1. Lines 292-298: If these calculations are not described in reference 40, the reference given for Blast, the authors should provide a reference (e.g. in either line 292 or at the use of the term PSSM matrix in line 289) to the source from which they learned (or could have learned) how to perform these calculations.

Ans: It is clarified in the revision that the process of creating the PSSM was taken from our previous work, and the corresponding reference was added in line 284-285.

  1. Line 304: Why did the authors choose Moon’s hydrophobicity index? Is this index the one typically used in analyzing protein/DNA interactions or is there another reason. I am not sure if it is necessary to include a parenthetical one phrase explanation of why they chose this index, but I suspect it some readers may wish to know the reason.

Ans:  In our previous works, we used different hydrophobicity indices but saw no effect on the performance of the corresponding algorithms. In the present work, we selected one of them, the Moon's hydrophobicity index, simply to be specific. 

  1. Unless I am missing something, the total number of features used to predict ΔΔG values is not explicitly stated until line 618. It should be stated somewhere in Section 2.4. I imagine that the machine learning techniques used in this paper are not as sensitive to the need for the number of datapoints to greatly exceed the number of independent variables as traditional regression techniques fitting one parameter per variable (possibly with an additional offset parameter as well as additional interaction parameters), and the cross-validation would detect if overfitting did occur as a result of insufficient degrees of freedom. However, I would imagine the feature vector still should have a lower dimensionality than the number of points in the training data set. Is this the case?

Ans: In both cases, the mutation occurring in protein or DNA, the number of features is less than the number of data points. This is now pointed out in the revised manuscript at the end of the method sections regarding protein mutation (section 2.4.1) and DNA mutation (section 2.4.2).

  1. Section 2.6 leaves the reader a bit unclear. Are the authors using a different approach than they did for the original version of SAMPDI-3D? If the approach is the same, it may be possible to abbreviate this section. If the approach is different, it should be noted. Is the code written so that users can customize which machine learning approach is used? Similar questions can be raised about Section 2.7, but so long as the answers are the same as for Section 2.6, there need not be any changes to Section 2.7.

Ans: The approach differs from the old SAMPDI-3D, which motivates us to describe it in detail.

  1. I thank the authors for providing the interesting analysis in Section 3.1. Has any similar analysis been performed previously? If so, the authors might want to reference it and compare their training set with previously analyzed datasets. If not, I do not think any changes are needed to this section.

Ans: We have done a similar analysis, but the goal was different (Int. J. Mol. Sci. 2023, 24(15), 12073; Curr Opin Struct Biol. 2023 Mar 23;80:102572). Previous works emphasized the type of mutations in the existing dataset and their relation to single nucleotide polymorphism (SNPs). In the current work, the emphasis is given to the difference between the representations of amino acids being mutated in two datasets, S496 (curated from ProNAB and literature used in SAMPDI-3D old version) and new dataset S177 (cleaned from ProNAB). This motivates the development of SAMPDI-3D version 2.

  1. Do the results presented in Section 3.2 indicate that the original SAMPDI-3D method was overfit on the S419 dataset? How did the other methods used in this section perform on that dataset? Relatedly, how did the other methods perform on the S596 dataset? The abstract of the paper claims that SAMPDI-3D v2 performs significantly better than other methods, but the only PCC values I see for other methods are obtained on the S177 dataset (Table 6). If the authors wish to claim significance, they should provide the PCC for the best performing other predictor (just as a number, preferably with some indication of its uncertainty) on the S596 dataset and compare it with the 0.65 ± 0.05 figure obtained from SAMPDI-3D v2. At the very least, the authors should compare the SAMPDI-3D v2 average correlation coefficient of 0.65 with the correlation coefficient for the best performing model on S596 using (although it is not perhaps a statistically rigorous use of this test) using Fisher r to z transformations. For what it’s worth, assuming that PremPDI remains the best performing model and the correlation coefficient on the S596 data for that method remains around 0.4, on a dataset with 596 datapoints, the z score for comparing a correlation coefficient of 0.4 with a correlation coefficient of 0.65 is > 6, which is highly significant by any standard.

Ans: Indeed, the referee is correct, and the results presented in Figures 2 and 3 indicate that the old version of SAMPDI-3D and the other existing methods are overfitted towards S419, and they perform poorly on the new data points (S177).  We can’t perform cross-validation on other methods as we did on the new version of SAMPDI-3D; instead, we checked the performance of methods over the new expanded dataset S596, and the results are shown in the supplementary material. We understand that this is not the best way to assess the performance, but it is done just to compare the new SAMPDI-3D and the existing prediction methods. The prediction using old SAMPDI-3D, PremPDI and mCSM-NA over the S596 dataset yielded PCC of 0.62, 0.44 and 0.45, respectively. The result showed that the PCC for the existing methods is lower than the PCC of 0.68 (best iteration) obtained from SAMPDI-3D version 2. Additionally, it has to be noted that cross-validation predictions during the training phase and final model predictions are different and should not be conflated. However, since we don’t have access to cross-validation results from the existing methods, we compared the cross-validation (an even stricter test as none of the data points overlapped with the training set while predicting for it) number from SAMPDI-3D version 2 with model prediction from existing methods. The PCC for model prediction for SAMPDI-3D version 2 on S596 is 0.95 after the model development, as is the case for other methods. 

We used Fisher r to z transformation to compare the PCCs from SAMPDI-3D, PremPDI and mCSM-NA to PCC from SAMPDI-3Dv2 (best iteration over cross-validation), which resulted in z-values 1.79 (two-tailed p-value 0.0735), 6.15 (two-tailed p-value <1.0e-4) and 5.93 (two-tailed p-value <1.0e-4). These results show significant improvement in PCC over cross-validation.

  1. In the Discussion section, the authors point out the importance of using accurate ΔΔG values to train their method and emphasize the importance of their database cleaning. However, it could be the case that, despite their careful cleaning, some of the ΔΔG values they used were inaccurate. Indeed, in the case of mutations with multiple associated ΔΔG values, it could be a rejected value that is correct. While the cross-validation performed would tend to support the overall accuracy of the model, I am curious as to whether the authors explored any of the cases where, during cross-validation, the model failed to accurately predict ΔΔG. Are there any cases in which cross-validation uncovered an error in the dataset used in this paper? In a similar vein, did the authors apply their method to rejected ΔΔG values to verify that they are indeed not what would be predicted from accepted ΔΔG values?

Ans: We have not observed evidence suggesting some experimental data points are wrong during the cross-validation. We would like to clarify that the number of multiple ΔΔGs reported for the same mutation is relatively small (less than 10% of the entries in S596), and even more, the multiple ΔΔGs were averaged but not rejected except for the outliers. 

  1. I am also curious (although it may be beyond the scope of this paper, and there may not be enough data to support this analysis) as to whether the success in prediction of ΔΔG values depends on the experiment used to obtain those ΔΔG values. Under the assumption that the predictions are overall accurate, do some experimental methods appear to give more accurate ΔΔG values than others?

Ans: The data points are very limited. If we would like to create a sub-dataset for each experimental method, it will be too small to provide insight into machine learning training and benchmarking.

Comments on the Quality of English Language

  1. In line 75, there should be a new paragraph beginning with “In addition to traditional methods” as the paragraph, as it currently stands, is very long.

Ans: A new paragraph has been added in line 74.

  1. Lines 76ff: “For example, protein binding microarrays (PBMs) is a fluorescence-based methods that assess the binding specificities of TFs to interacting DNA”. I am not sure if PBMs is a particular method or not, which impacts whether the verb form used should be “is” or “are”. I suspect “methods” needs to be singular (“method”) and perhaps the sentence should just be rephrased to sound less awkward no matter what the correct number agreement is.

Ans: Protein Binding Microarrays (PBMs) is a particular method, and the sentence has been updated to reflect this using singular nouns and verbs as suggested. Also, TFs has been written as transcription factors in line 77.

  1. Line 86: “it’s” should be “its”.

Ans: The sentence been slightly modified as ‘the protein and the interacting DNA’ in line 85-86.

  1. The paragraph beginning on line 101 is also very long and perhaps should be split up. The obvious place (to me, at least) to split is it to begin a new paragraph with “Other models available” (splitting the authors’ approaches from other approaches) on line 111. This splits the paragraph a wee bit more asymmetrically than would be desirable, however.

Ans: The referee’s point is well taken, and the paragraph has been split at line 110, beginning with ‘mCSM-NA [31] and mmCSM-NA [32]….” which is slightly modified from the previous form.

  1. Line 145f: “no other methods can predict the effects of mutation in DNA on ΔΔG” is a bit strong. Perhaps “none of the other methods mentioned above can predict the effects of DNA mutation on ΔΔG” would be better?

Ans: The line has been updated to make it less strong, as the referee suggested in line 142-143.

  1. Line 161: Should be “Results section” (not sure if section should be capitalized or not) as opposed to “result section”.

Ans: ‘result section’ has been changed to ‘results section’ in line 153.

  1. Lines 168-170: “Since our model development requires experimentally measured changes of binding free energy (ΔΔG) upon mutation of either in protein or DNA” is a sentence fragment.

Ans: Yes, it was erroneously a sentence fragment and has been corrected in line 166-167 with a slight modification in sentence itself.

  1. Line 227f: the phrase “Additionally, 34 data points had issues with the PDB components (e.g., missing residues), and they were removed,” can be shortened to “Additionally, we removed 34 data points that had issues with the PDB components (e.g., missing residues),”. Another possible change (to eliminate a couple of words) is found on line 240f: “In our previous version of SAMPDI-3D, the S419 dataset was used for training and testing of ΔΔG caused by protein mutation.”, which can be edited to “Our previous version of SAMPDI-3D used the S419 dataset for training and testing of ΔΔG caused by protein mutation.” The authors should double check that they are as succinct as possible to keep the paper a manageable size should they have to add additional material in response to reviewers.

Ans: The phrase “Additionally, 34 data points had issues with the PDB components (e.g., missing residues), and they were removed,” has been updated to shorten it as mentioned by the reviewer in line 224-225. The same was done for line 240, now in line 237-238.

  1. Line 261f: I see that I am not the only one who uses parentheticals such as “(ref our paper)” in the drafting process and then forgets to actually replace them with the actual reference in the submitted manuscript. Of course, this should be fixed during the revision.

Ans: This was a glaring mistake and has been corrected in lines 257.

  1. Lines 652ff: The sentence “The work resulted in a new version of the SAMPDI-3D method, which is now more accurate than the previous version, equipped with new features to strengthen the specificity of the predictions, especially for mutations in the DNA and trained on a larger database.” is long and awkward, and could use revision.

    Ans: The sentence has been revised and split into two for clarity in line 663-664.

Round 2

Reviewer 2 Report

Comments and Suggestions for Authors

I thank the authors for their extensive reviews in response to reviewer comments. These well-thought-out changes address most of my concerns with this paper.

While the authors have clarified the differences between SAMPDI-3D and SAMPDI-3Dv2 in their responses to my queries and have largely clarified the distinctive aspects of SAMPDI-3Dv2 in their revisions, it is still unclear to me how extensible the code for SAMPDI-3Dv2 is. The authors would do well to indicate in section 2.6 that they used a different approach for machine learning in SAMPDI-3Dv2 than in SAMPDI-3D.

Additionally, as I understand it, users can train the authors’ method locally. If this is the case, the authors should indicate whether or such local training is restricted to the “optimal machine learning approach” described in this paper or if the user can select which machine learning approach is used when running the authors’ software locally. This can be clarified in either section 2.6 itself or in section 3.4, where the authors describe the implementation of their method.

Similarly, the authors should indicate whether users (of the local software) are restricted to the authors’ choice of hyperparameters or whether they can choose their own values for these parameters.

Author Response

We thank again the reviewer for useful comments which are addressed below:

1) While the authors have clarified the differences between SAMPDI-3D and SAMPDI-3Dv2 in their responses to my queries and have largely clarified the distinctive aspects of SAMPDI-3Dv2 in their revisions, it is still unclear to me how extensible the code for SAMPDI-3Dv2 is. The authors would do well to indicate in section 2.6 that they used a different approach for machine learning in SAMPDI-3Dv2 than in SAMPDI-3D.

Ans: The difference between the methodology of the old SAMPDI-3D and the new SAMPDI-3Dv2 is outlined in 2.6, lines 491-496. It is pointed out that the new version of SAMPDI-3D, v2, investigated the performance of various regression methods to select the best one, which was not done in the old development. Regarding extensibility of SAMPDI-3Dv2 code, it is available for download and the users can edit it and incorporate new features and other approaches.

2) Additionally, as I understand it, users can train the authors’ method locally. If this is the case, the authors should indicate whether or such local training is restricted to the “optimal machine learning approach” described in this paper or if the user can select which machine learning approach is used when running the authors’ software locally. This can be clarified in either section 2.6 itself or in section 3.4, where the authors describe the implementation of their method.

Ans: The downloadable and web implemented SAMPDI-3Dv2 is already trained code and thus cannot be subjected to further training. However, the users can use the code, make modifications and with some extra work come up with retrained software. This is stated in the revision in 3.4, line 662-663, where we indicate that the users can use local code to make predictions and modifications if needed.

3) Similarly, the authors should indicate whether users (of the local software) are restricted to the authors’ choice of hyperparameters or whether they can choose their own values for these parameters.

Ans: As indicated above, we provide trained model for download where the hyperparameters are already selected. However, the users are free to make modifications as they want.